# Prediction of Surgical Outcome by Tumor Volume Doubling Time via Stereo Imaging Software in Early Non-Small Cell Lung Cancer

**DOI:** 10.3390/cancers15153952

**Published:** 2023-08-03

**Authors:** Chia-Chi Liu, Ya-Fu Cheng, Pei-Cing Ke, Yi-Ling Chen, Ching-Min Lin, Bing-Yen Wang

**Affiliations:** 1Division of Thoracic Surgery, Department of Surgery, Changhua Christian Hospital, Changhua 50006, Taiwan; 182931@cch.org.tw (C.-C.L.); 181033@cch.org.tw (Y.-F.C.); 180126@cch.org.tw (P.-C.K.); 182907@cch.org.tw (C.-M.L.); 2Surgery Clinical Research Center, Changhua Christian Hospital, Changhua 50006, Taiwan; 181190@cch.org.tw; 3Department of Post-Baccalaureate Medicine, College of Medicine, National Chung Hsing University, Taichung 40227, Taiwan; 4School of Medicine, Chung Shan Medical University, Taichung 40201, Taiwan; 5School of Medicine, College of Medicine, Kaohsiung Medical University, Kaohsiung 80756, Taiwan; 6Institute of Genomics and Bioinformatics, National Chung Hsing University, Taichung 40227, Taiwan; 7Center for General Education, Ming Dao University, Changhua 52345, Taiwan

**Keywords:** non-small cell lung cancer, segmentectomy, survival rate, volume doubling time, wedge resection

## Abstract

**Simple Summary:**

We aimed to investigate if VDT could be applied as a predictor of clinical outcome in segmentectomy and wedge resection. We retrospectively studied 96 NSCLC patients post sublobar resection from 2012 to 2018, collecting two chest CT scans preoperatively of each case and calculating VDT. The receiver operating characteristic curve was constructed to identify the optimal cut-off point of VDTs as 133 days. We divided patients into two groups: VDT < 133 days (n = 22) and VDT ≥ 133 days (n = 74). Univariable and multivariable analyses were performed for comparative purposes. Our study demonstrated that the five year OS rates of patients with VDTs ≧ 133 days and VDTs < 133 days, respectively, were 89.9% and 71.9% (*p* = 0.003), and the five year DFS rates were 95.9% and 61.5% (*p* = 0.002). Thus, we concluded that VDT can be a powerful prognostic predictor and provides an essential role in planning surgical procedures.

**Abstract:**

Background: Volume doubling time (VDT) has been proven to be a powerful predictor of lung cancer progression. In non-small cell lung cancer patients receiving sublobar resection, the discussion of correlation between VDT and surgery was absent. We proposed to investigate the surgical outcomes according to VDT. Methods: We retrospectively studied 96 cases post sublobar resection from 2012 to 2018, collecting two chest CT scans preoperatively of each case and calculating the VDT. The receiver operating characteristic curve was constructed to identify the optimal cut-off point of VDTs as 133 days. We divided patients into two groups: VDT < 133 days and VDT ≥ 133 days. Univariable and multivariable analyses were performed for comparative purposes. Results: Univariable and multivariable analyses revealed that the consolidation and tumor diameter ratio was the factor of overall survival (OS), and VDT was the only factor of disease-free survival (DFS). The five year OS rates of patients with VDTs ≥ 133 days and VDTs < 133 days, respectively, were 89.9% and 71.9%, and the five year DFS rates were 95.9% and 61.5%. Conclusion: As VDT serves as a powerful prognostic predictor and provides an essential role in planning surgical procedures, the evaluation of VDT preoperatively is highly suggested.

## 1. Introduction

Among all cancers, lung cancer is a major cause of cancer death worldwide, accounting for 18.4% of all cancer deaths in 2018 [1,2]. Non-small cell lung cancer (NSCLC) accounts for about 85% to 90% of all types of lung cancer [3]. Recently, volume doubling time (VDT) in lung cancer screening has been proven to be helpful for distinguishing high-risk lung nodules from low-risk ones. The nodule management strategy of the Dutch–Belgian NELSON trial is based on volume and VDT assessment [4]. Some studies have reported that slow-growing tumors have a VDT of longer than 400 days; therefore, 400 days is considered an optimal cut-off time to determine malignant lesions from benign ones [4,5,6].

VDT has been a predictor of the prognosis of lung cancer progression, associated with factors such as consolidation and tumor diameter ratio (C/D ratio), smoking history, underlying chronic obstructive pulmonary disease (COPD), epidermal growth factor receptor (EGFR) mutation, forced expiratory volume in one second (FEV1), tumor stage and subtype. As for surgical approach, the indications for video-assisted thoracic surgery (VATS) sublobar resections in NSCLC were described [7]. According to the National Comprehensive Cancer Network Clinical Practice Guidelines in Oncology (NCCN Guidelines) for NSCLC [8], sublobar resection is appropriate for patients who are contraindicated for lobectomy and for those with a peripheral nodule ≤ 2 cm with radiologic surveillance confirming a doubling time ≥ 400 days. However, the correlation between VDT and sublobar resection in postoperative outcomes remains unclear.

The purpose of this study is to further investigate if VDT could be applied as a predictor of clinical outcome in segmentectomy and wedge resection. We also compared overall and disease-free survival rates between segmentectomy and wedge resection under the determination by VDT to identify if different outcomes could be found, which could lead to more specific indications for either technique.

## 2. Materials and Methods

### 2.1. Patients

Our study was a retrospective analysis in our institute (Changhua Christian Hospital, Changhua, Taiwan) and was approved by our institutional review board (IRB-210124). Informed consent from all participants was waived. A search of the database of our institution identified 485 patients with clinical stage IA NSCLC (classified using the 8th edition of the AJCC Cancer Staging Manual [9]), which was defined as tumor size ≤ 2 cm in diameter, from January 2012 to December 2018. Exclusion criteria for the study were as follows: (1) patients who underwent incomplete resection, (2) patients who received neoadjuvant therapy including chemotherapy, radiotherapy, target therapy and immunotherapy, (3) there was only one preoperative CT or two preoperative CT scans that were <30 days apart, (4) more than one tumor was diagnosed at the same time, (5) there were tumors invading the chest wall, mediastinum structures, distal metastasis, or positive clinical lymph node metastasis, (6) incomplete data or loss of follow-up, (7) an extreme VDT of more than five years. At last, we excluded those receiving lobectomy since we aimed to inquire into the role of VDT in sublobar resection only, and there were relatively few such patients. The remaining 96 patients were analyzed as subjects of this study. The tumor, node and stage were determined in accordance with the 8th edition of TNM classification of malignant tumors.

### 2.2. Surgery

If a lung lesion was detected on a chest X-ray or computerized tomography (CT) scan, we would arrange a further CT-guided or endobronchial ultrasound biopsy to confirm the pathological diagnosis. Subsequently, we would schedule an abdominal sonography, brain magnetic resonance imaging, bone scans and positron emission tomography scans of the chest for routine lung cancer staging work-up. Based on preoperative image evaluation, we would opt for a simple wedge resection if the lung nodules were located in the peripheral lung (outer third). After wedge resection, we would check the margin intraoperatively at back table to confirm if the margins from pleura were less than 2 cm or less the diameter of the nodule. We would reconstruct three-dimensional images preoperatively and perform segmentectomy once the lesion located centrally or a safe margin could not be confirmed. All the patients underwent video-assisted thoracic surgery. Under general anesthesia, we performed double-lumen endotracheal tube intubation for one-lung ventilation. Subsequently, we positioned the patient in the lateral decubitus position, and the operative field was prepped. For either segmentectomy or wedge resection, we routinely conducted single-port thoracoscopic surgery. In the case of segmentectomy, we utilized commercial software to preoperatively reconstruct a three-dimensional image for simulation. This three-dimensional model allowed us to identify the target segment, vessels and bronchus accurately. Using a scalpel, we isolated the vessels and bronchus, followed by division using staplers. By following the line demarcated with the inflation and deflation method, we successfully divided the targeted segment. For wedge resection, we performed the simulation and localization of the target nodular lesion in the hybrid operating room. To aid in localization, we employed CT guidance and injected methylene blue to dye the tissue surrounding the nodule. Subsequently, under thoracoscopic guidance, we identified the dye point and conducted wedge resection with staplers, ensuring a safety margin was maintained. We measured all the margins on the specimen intraoperatively at the back table after removing the lesion. If any margins were found to be <2 cm or smaller than the tumor size, we performed extended resection until a safe margin was achieved and confirmed. Additionally, if the safe margin was found to be insufficient after wide wedge resection, we would convert the surgical method to segmentectomy or lobectomy. Once the intraoperative frozen section showed positive malignant finding, including carcinoma in situ, the lymphadenectomy of subcarinal, subaortic, paraoesophageal and the lymph nodes near to the lung lesion would be performed for further staging. In comparison with segmentectomy, lobectomy should remain the standard procedure when N1 disease is present [10]. We performed lymph node dissection with frozen section examination when lymph node metastases were suspected or when the lymph node size was more than 1 cm. If the frozen section showed a positive malignant finding, we would proceed with the lobectomy, and those cases would be excluded from our study.

### 2.3. Patient Follow-Up

Patients were examined at 1 week and 1 month after discharge, then at 3 month intervals for the first year, at 6 month intervals for the second year, and at 1 year intervals thereafter. The follow-up evaluation included a physical examination, chest radiography, blood examination and CT scan of the chest. Further evaluations, including brain magnetic resonance imaging, bone scintigraphy and positron emission tomography scans, were performed when symptoms or signs of recurrence were detected. We followed the recurrence via diagnostic imaging of the lesions on an outpatient basis until the end of 2020. The recurrence time was defined as the date of identification based radiological findings, which was followed by pathological confirmation. The standard definition of local and regional recurrence does not currently exist, leading to variation in recurrence rates based on the different definitions used [11]. In our medical center, we have privately defined local recurrence to involve the bronchial stump, ipsilateral hilum, interlobar staple line, ipsilateral mediastinal lymph nodes, ipsilateral lung parenchyma and pleural effusion in the ipsilateral thoracic cavity. On the other hand, we defined regional recurrence as the involvement of contralateral mediastinal/hilum lymph nodes, supraclavicular lymph nodes, contralateral lung parenchyma and the chest wall. Among seven recurrent cases, there was one general relapse with distant metastasis and six local or regional relapses.

### 2.4. Clinical Features of Patients

Clinical characteristics of the entire study population, including age, sex, C/D ratio, smoking history, underlying COPD, the presence of an EGFR genetic mutation, FEV1, tumor staging, tumor location and subtype of lung cancer, were recorded.

### 2.5. Analysis of VDTs

For each patient, two chest CT examinations were selected: the first CT scan that showed a visible lesion and the last CT scan that was performed before the surgery. The CT scan slice thickness ranged from 1.0 mm to 5.0 mm, most of them were 2.5 mm. Tumor segmentation and three-dimensional reconstruction were performed independently on the two CT scans from each patient using commercial software (Ziostation2; Ziosoft, Tokyo, Japan). For each tumor, the software automatically outlined the tumor on the axial CT images slice by slice. After tumor segmentation, the software reconstructed and calculated the volume of each segmented tumor. Then, the VDT was calculated by using the following equation: VDT = (T ∙ log^2^)/[log (V1/V0)], where V0 is the tumor’s volume on the first CT image, V1 is the tumor’s volume on the last CT image and T is defined as the interval between the two CT scans in days (Figure 1). A limitation of our study was the inability to obtain every CT scan with a 1 mm slice thickness. Our data showed that the CT slice thickness varied from 1 mm to 5 mm, with 2.5 mm being the most common thickness in our data. The reproducibility of radiomic features in lung cancer is indeed affected by CT slice thickness [12]. However, a 2.5 mm slice thickness might still be acceptable for three-dimensional reconstruction with some technique modifications [13,14]. Additionally, it could be applied as preoperative simulation in thoracic surgery for early-stage NSCLC [15].

### 2.6. Statistical Analysis

Overall survival (OS) was measured from the date of surgery to the date of death from any cause or the date of the last follow-up visit before 2021. Disease-free survival (DFS) was measured from the date of the surgery until the first detection of a chest lesion on an image, which was confirmed pathologically afterward. All cumulative OS and DFS rates were estimated by Kaplan–Meier curves, and differences in variables were determined using the log-rank test. A receiver operating characteristic (ROC) curve was constructed, and we used the area under the curve to identify the optimal VDT cut-off point as 133 days for the differentiation of the better survival outcomes from the poor ones. The baseline characteristics were compared between patients using a Mann–Whitney U test for continuous variables and a chi-squared test for categorical variables. Univariable and multivariable analyses for OS and DFS were performed using a logistic regression model and presented via odds ratio. A *p* value less than 0.05 was considered to indicate statistical significance. All statistical analyses were performed using SPSS statistical software (SPSS package, version 23.0; SPSS, Chicago, IL, USA).

## 3. Results

### 3.1. Baseline Characteristics

The study included 96 patients (mean age: 59.95 ± 11.38 years) with the baseline characteristics shown in the table (Table 1). We performed the ROC curve method to determine an optimal VDT cut-off point of 133 days, and we divided all patients into two groups based on better or poor prognosis. There were significantly more male patients with shorter VDTs. The mean tumor size was 0.96 ± 0.42 cm; it was not associated with VDT. Factors that had been mentioned in previous studies, which were related to different VDTs, were included in our analysis, such as C/D ratio, smoking history, COPD, EGFR genetic mutation, FEV1, histological types and subtypes. These factors did show differences between the two VDT groups in our data. Sublobar resection was proven to be no inferior to lobectomy in terms of disease-free survival for patients with peripheral NSCLC, with a tumor size of 2 cm or less and pathologically confirmed node-negative disease in the hilar and mediastinal lymph nodes [16,17]. For early-stage NCSLC patients, wedge resection was proved to be a safe and feasible sublobar resection method, which was equivalent and comparable to segmentectomy and lobectomy in selective cases [18,19,20]. The patients in our study underwent sublobar resection, either by segmentectomy (n = 45) or wedge resection (n = 51). There was no statistically significant difference in VDT between the two procedures.

### 3.2. Univariable and Multivariable Analyses

A univariable analysis was conducted to identify significant prognostic factors for OS (Table 2). Among these factors, age, sex, smoking history, COPD, C/D ratio and VDT were found to have an impact on OS. However, after performing a multivariable analysis separately, none of these factors were found to be independently associated with OS. Nevertheless, based on the odds ratio, COPD and VDT appeared to have a potential correlation with OS. Moving on, we also conducted univariable and multivariable analyses for disease-free survival (DFS) (Table 3). As a result, factors such as smoking history, COPD, C/D ratio and VDT were observed to potentially influence DFS. However, VDT was identified as the only significant factor affecting DFS (*p* = 0.008, odds ratio = 55.63).

### 3.3. Overall Survival and Disease-Free Survival after Sublobar Resection in Different VDT Groups

The five year OS rates of patients with VDTs ≥ 133 days and VDTs < 133 days, respectively, were 89.9% and 71.9% (Figure 2A, *p* = 0.003), and the five year DFS rates of patients with VDTs ≥ 133 days and VDTs < 133 days, respectively, were 95.9% and 61.5% (Figure 2B, *p* = 0.002). In the group with VDTs ≥ 133 days, the five year OS rates of patients who underwent segmentectomy and wedge resection, respectively, were 95.0% and 84.7% (Figure 3(A-1)); the five year DFS rates were 100% and 92.6% (Figure 3(A-2)). In the group with VDTs < 133 days, the five year OS rates of patients who underwent segmentectomy and wedge resection, respectively, were 74.1% and 70.0% (Figure 3(B-1)); the five year DFS rates were 70.1% and 58.3% (Figure 3(B-2)).

## 4. Discussion

In this current study, we determined OS and DFS differed significantly between patients with VDTs ≥ 133 days and patients with VDTs < 133 days. A VDT (<133 days) could be a preoperative predictor of surgical outcome in early NSCLC patients.

In the clinical characteristics of our series, we found the majority were female and a vast majority were non-smokers. This trend had been discovered by recent studies, which revealed high rates of lung cancer among most never-smoking Asian female ethnic groups [21,22]. There have been many studies that found that VDT is associated with factors correlating to lung cancer and can be used as a predictor of prognosis [23]. In addition to sex and smoking history, C/D ratio, COPD, EGFR mutation and tumor stage and location were statistically significantly different between the different VDT groups in our data.

We used an ROC curve analysis to identify an optimal VDT cut-off point. Setojima et al. included 258 NSCLC patients from January 2012 to December 2015. Using a method similar to ours, they obtained solid-part tumor volume doubling times and concluded that there was a significantly higher five year recurrence-free survival rate in the group with VDTs > 215 days [24]. However, they did not include those who underwent segmentectomy or wedge resection. Thus, different inclusion criteria for patients may cause different VDT cut-off points. Our study did not point out which cutoff doubling time was the most reliable, but we had proved that the same statistical method in analysis of the relationship between different objects and VDT is available.

As for different tumor cell types and subtypes, there are some studies that have analyzed their relationship to VDT. A recent study from Hong et al. in 2020 included 172 patients with surgically resected lung adenocarcinoma [25]. They evaluated the VDTs via three-dimensional semiautomatic segmentation, and they determined subtypes according to the 2011 World Health Organization histologic classification. In addition, they compared the VDTs of the subtypes to the VDT of squamous cell carcinoma (SqCC). Their results showed a significantly shorter VDT for solid/micropapillary predominant tumors among all the subtypes and a significantly shorter VDT for SqCC than any of the subtypes. Similar results were also found in previous studies [26,27,28]. Our data showed a significant difference in VDT between SqCC and adenocarcinoma in situ (AIS) patients, but the further univariable and multivariable analyses showed no effect on our conclusion.

In recent years, thoracic surgical resection has been a significantly effective and beneficial treatment of choice for patients with early-stage NSCLC [29]. As for patients with resected stage I NSCLC, the five year survival rate ranges from 44% to 72% [30,31,32]. Compared to lobectomy, sublobar resections, including segmentectomy and wedge resection, are particularly indicated in older people and in patients with a considerable risk of comorbidity or reduced respiratory functional reserve, and they seem to result in better DFS [33,34]. However, controversies still exist [35]. Our study excluded lobectomy cases in order to avoid these controversies.

Regarding the correlation between VDT and surgical methods, Miura et al. evaluated 231 NSCLC patients grouped by lobectomy (n = 206), segmentectomy (n = 24) and pneumonectomy (n = 1), and they found no significant difference according to VDT (<400 days and ≥400 days) [36]. In our study, the median VDT of the wedge resection group was 338.60 days, which was not statistically significantly different from the 366.90 days for the segmentectomy group. Furthermore, we analyzed the OS and DFS of sublobar resections according to VDT (<133 days and ≥133 days); there were no significant differences of surgical outcomes between segmentectomies and wedge resections. Thus, we concluded that both segmentectomy and wedge resection can be performed on early-stage NSCLC patients.

There were several limitations of our study. First, the data are derived from a single medical center. Second, the retrospective design may lead to selective bias. Third, the study period was limited to data available from 2012 to 2018 only, which could result in inaccuracies in OS and DFS due to inadequate observations and shortness of long-term follow-up. Fourth, the slice thickness of the CT scan varied among the patients, which may have affected the volume measurements and caused discrepancies, resulting in inaccurate calculations of VDTs. Using a 1 mm slice thickness of the CT scan for the analysis of VDT can lead to higher accuracy compared to using 2.5 mm or larger slices, and the threshold value of 133 days may differ accordingly. The new data can provide higher reliability as a predictive factor for prognosis.

## 5. Conclusions

Despite the limitations, our study has proposed a research method aimed at improving surgical outcomes. In conclusion, as the surgical outcomes of early-stage NSCLC patients are significantly different according to VDT, volume doubling time serves as a powerful prognostic predictor and plays an important role in planning surgical procedures. Thus, preoperatively calculating VDT is highly suggested. In the future, a larger cohort and more meticulous imaging will be necessary to obtain a more precise VDT for a more powerful prediction. Due to our small and limited sample size, we were unable to definitively determine the performance of wedge resection or segmentectomy on VDT. Further multicenter, larger sample, prospective and randomized studies are necessary to confirm the results of this study.

## Figures and Tables

**Figure 1 cancers-15-03952-f001:**
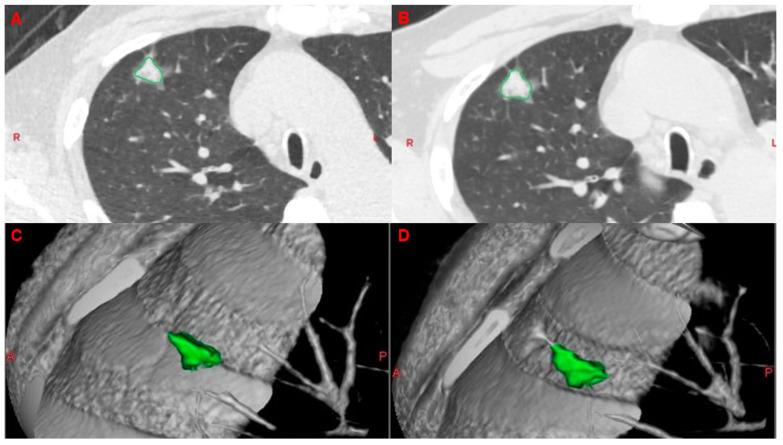
Images in a 66-year-old man with a solid right upper lobe lung lesion. (**A**) The initial axial CT scan shows a lung nodule with a maximal diameter of 17.8 mm. (**B**) On the axial CT scan obtained after 1.5 months, the lung nodule was larger, with a maximal diameter of 19.5 mm. (**C**) The initial tumor was segmented three dimensionally and its volume was calculated as 925.4 mm^3^. (**D**) The tumor volume increased to 1271.8 mm^3^. The volume doubling time was 101.2 days.

**Figure 2 cancers-15-03952-f002:**
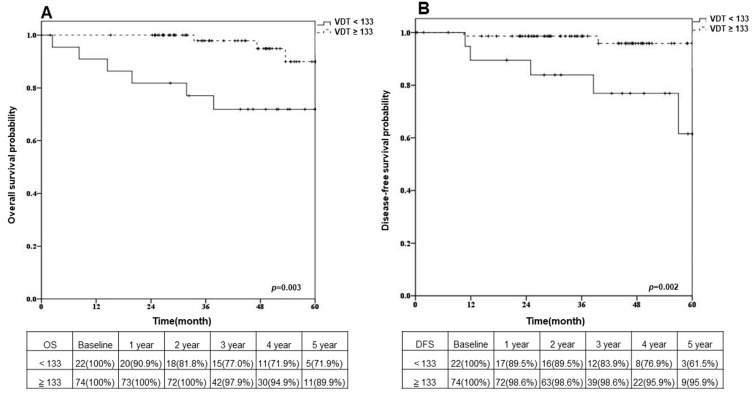
Kaplan–Meier curves stratified by VDT (<133 days and ≥133 days) are plotted for (**A**) overall survival probability and (**B**) disease-free survival probability. *p* values were acquired by using the log-rank test.

**Figure 3 cancers-15-03952-f003:**
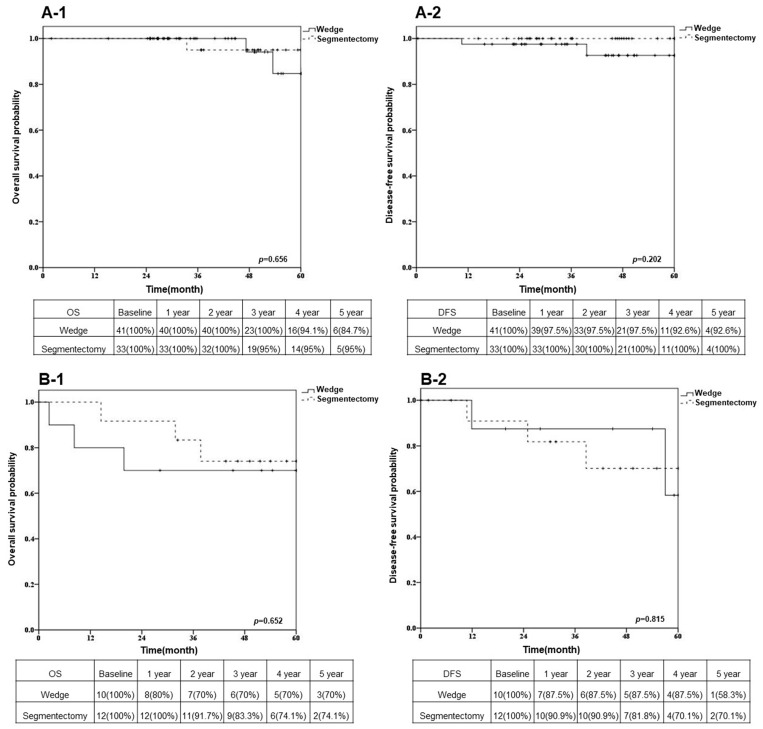
In the group with VDT ≥ 133 days, Kaplan–Meier curves stratified by surgical method are shown for (**A-1**) overall survival and (**A-2**) disease-free survival. The same analysis was presented in the group with VDT < 133 days, with (**B-1**) for overall survival and (**B-2**) for disease-free survival.

**Table 1 cancers-15-03952-t001:** Clinical characteristics of the patients.

Factors (Mean ± SD)	All Patients	VDT ≥ 133 Days	VDT < 133 Days	*p* Value
Number of patients	96	74	22	
Age (years) (Mean ± SD)	59.95 ± 11.38	59.53 ± 11.44	61.36 ± 11.35	0.36
Gender				
Male	39 (40.6%)	26 (35.1%)	13 (59.1%)	0.01
Female	57 (59.4%)	48 (64.9%)	9 (40.9%)	0.02
Tumor size (cm) (Mean ± SD)	0.96 ± 0.42	0.93 ± 0.40	1.06 ± 0.46	0.18
C/D ratio (%) (Mean ± SD)	11.67 ± 24.33	8.78 ± 22.19	21.41 ± 28.96	0.009
Smoking				
Ever	16 (16.7%)	8 (10.8%)	8 (36.4%)	<0.001
Never	80 (83.3%)	66 (89.2%)	14 (63.6%)	0.04
COPD				
Yes	9 (9.4%)	5 (6.8%)	4 (18.2%)	0.03
No	87 (90.6%)	69 (93.2%)	18 (81.8%)	0.41
EGFR mutation				
Positive	10 (10.4%)	4 (5.4%)	6 (27.3%)	<0.001
Negative	86 (89.6%)	70 (94.6%)	16 (72.7%)	0.09
FEV1(%) (Mean ± SD)	92.35 ± 16.78	93.79 ± 15.82	87.44 ± 19.36	0.53
T stage				
Tis	29 (30.2%)	28 (37.8%)	1 (4.5%)	<0.001
T1	62 (64.6%)	45 (60.8%)	17 (77.3%)	0.17
T2	5 (5.2%)	1 (1.4%)	4 (18.2%)	<0.001
TNM stage				
Stage 0	29 (30.2%)	28 (37.8%)	1 (4.5%)	<0.001
Stage I	67 (69.8%)	46 (62.2%)	21 (95.5%)	0.007
Tumor location				
Left	44 (45.8%)	36 (48.6%)	8 (36.4%)	0.16
Right	52 (54.2%)	38 (51.4%)	14 (63.6%)	0.23
Histological type				
Adenocarcinoma	55 (57.3%)	40 (54.1%)	15 (68.2%)	0.21
SqCC	7 (7.3%)	2 (2.7%)	5 (22.8%)	<0.001
AIS	21 (21.9%)	21 (28.3%)	0 (0.0%)	<0.001
MIA	10 (10.4%)	9 (12.2%)	1 (4.5%)	0.09
Other	3 (3.1%)	2 (2.7%)	1 (4.5%)	0.48
Subtypes				
Lepidic	5 (9.1%)	3 (7.5%)	2 (13.3%)	0.275
Non-lepidic	50 (90.9%)	37 (92.5%)	13 (86.7%)	0.655
Surgical method				
Segmentectomy	45 (46.9%)	33 (44.6%)	12 (54.5%)	0.32
Wedge resection	51 (53.1%)	41 (55.4%)	10 (45.5%)	0.37

AIS = adenocarcinoma in situ; C/D ratio = consolidation and tumor diameter ratio; COPD = chronic obstructive pulmonary disease; EGFR = epidermal growth factor receptor; FEV1 = forced expiratory volume in the first second; MIA = minimally invasive adenocarcinoma; SqCC = squamous cell carcinoma; Tis = tumor in situ.

**Table 2 cancers-15-03952-t002:** Univariable and multivariable analyses of factors affecting overall survival.

	Univariate		Multivariate	
	Odds Ratio	*p* Value	Odds Ratio	*p* Value
Age (years)	1.10	0.009	1.11	0.17
Gender				
Male	14.45	0.01	2.59	0.59
Female	1			
Smoking				
Yes	15.40	<0.001	0.78	0.88
No	1			
COPD				
Yes	13.12	0.002	69.45	0.09
No	1			
EGFR mutation				
Positive	1.08	0.94	0.26	0.49
Negative	1			
FEV1(%)	−0.08	0.48	1.03	0.47
TNM stage				
Stage 0	1			
Stage I	2.51 × 10^8^	0.99	2.88 × 10^7^	0.99
Tumor location				
Left	0.94	0.93	0.23	0.38
Right	1			
C/D ratio				
≧50%	14.29	0.001	7.16	0.22
<50%	1			
Surgery				
Wedge resection	1			
Segmentectomy	0.90	0.88	0.23	0.35
VDT				
≥133 days	1			
<133 days	8.88	0.004	23.93	0.07

C/D ratio = consolidation and tumor diameter ratio; COPD = chronic obstructive pulmonary disease; EGFR = epidermal growth factor receptor; FEV1 = forced expiratory volume in the first second.

**Table 3 cancers-15-03952-t003:** Univariable and multivariable analyses of factors affecting disease-free survival.

	Univariate		Multivariate	
	Odds Ratio	*p* Value	Odds Ratio	*p* Value
Age (years)	0.99	0.74	0.93	0.21
Gender				
Male	2.06	0.36	2.07	0.54
Female	1			
Smoking				
Yes	0.82	0.86	0.003	0.05
No	1			
COPD				
Yes	1.69	0.65	262.29	0.09
No	1			
EGFR mutation				
Positive	1.48	0.73	0.29	0.40
Negative	1			
FEV1(%)	1.00	0.89	1.01	0.73
TNM stage				
Stage 0	1			
Stage I	2.75	0.36	0.75	0.85
Tumor location				
Left	1			
Right	2.23	0.35	0.35	0.34
C/D ratio				
≧50%	1			
<50%	3.16	0.20	24.19	0.07
Surgery				
Wedge resection	1			
Segmentectomy	0.84	0.83	0.13	0.17
VDT				
≥133 days	1			
<133 days	10.59	0.007	55.63	0.008

C/D ratio = consolidation and tumor diameter ratio; COPD = chronic obstructive pulmonary disease; EGFR = epidermal growth factor receptor; FEV1 = forced expiratory volume in the first second.

## Data Availability

Not applicable.

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
