# Peer review of "Prediction of Surgical Outcome by Tumor Volume Doubling Time via Stereo Imaging Software in Early Non-Small Cell Lung Cancer"

_cancers, 2023, doi:10.3390/cancers15153952_

Round 1

Reviewer 1 Report

Thank you for the opportunity to analyze your article.       

In this article, authors aimed to determine if the volume doubling time (VDT) could be a predictor of clinical outcomes in sublobar resection. 

            Concerning the introduction:

            The introduction is well written with good references.

First comment about the inclusion of wedge resections… A wedge is not an anatomical lung resection.

            Concerning the methodology:

            Population:

            You didn’t mention brain imaging in the preoperative imaging assessment, did you perform it for early-stage NSCLC?   

            Surgery:

            The description of the surgical procedure needs to be more detailed. 

            Did you plan the segmentectomy with 3D reconstructions? For the latest cases for example?
            How did you choose to perform a segmentectomy or a wedge?

            Did you analyze oncological margins on frozen section?

            Did you analyze N1 lymph nodes on frozen section in order to extend the resection in case of N1 positive nodes? 

            Because performing a wedge resection didn’t allow to perform a lymph node assessment of area 12 and sometimes 11, why did you choose to include those patients in your work? Why not include lobectomy for early stage? 

            Follow up:

Can you define local and regional relapse please? 

Analysis of VDT:

How did you consider solid part and ground grass opacity part for the VDT? Because we are talking about early-stage NSCLC, is 2.5mm slice thickness a good criteria? 1mm isn’t better? Because the tumor size is 1cm mean in the results table. 

Concerning the statistical analysis conducted: 

            No major concerns about it. 

            Concerning the results:

            Results are well reported and clearly presented, but:

-       I’m still astonished to see so much wedge resections performed for “fit patient” with a good FEV1.

-       Need to report lymph node status with more details.

-       More than the VDT the quality of the surgery is a strong prognostic factor for survival. 

-       Concerning the tumor size in the Table 1, please precise mean.

-       Concerning the analysis made, the kind of surgical resection may have induced a bias, not statistically significant in the multivariate analysis but there is a trend. Line 180 and 181, be more cautious when you didn’t report any difference concerning 5-year OS and DFS between segmentectomy and wedges. 

            Concerning the discussion:

It’s a well written discussion well documented with good references. 

But I’m still embarrassed with the lack of data dealing with surgical procedure moreover concerning wedges resections.

References dealing with histologic subtypes are interesting but you can’t apply this to your population unfortunately. 

The 133 days threshold is a powerful threshold according your date, but may be impacted by the slice thickness of CT scans for small lesion of mean 1cm. Can it be different with slice thickness of 1mm? As we have today with modern imaging or 0.8mm? 

Nevertheless, limitations of your work are described. 

Concerning the conclusion:

Need to give more details about your results with your protocol, because there are some biases. 

            Nevertheless, it’s a well written, easy reading and interesting article, that need some precisions.

            Congratulations to authors for this work. 

Reviewer 2 Report

The study aimed to investigate the relationship between VDT and surgical outcomes in early-stage NSCLC patients. The results indicated that there were significant differences in OS and DFS between patients with VDTs ≥ 133 days and those with VDTs < 133 days. ROC curve analysis was used to identify the optimal VDT cut-off point, which was found to be 133 days. The calculation method for the cut- off point should be written in more detail, and it should be noted that the study has some limitations, such as the retrospective design and the single-center setting, which may affect the generalizability of the findings. Additionally, the variation in CT scan slice thickness among patients may have affected the accuracy of the volume measurements and VDT calculations.

Moreover, odds ratios were presented in the multivariable analysis without clear interpretation. It would be helpful to provide a more detailed explanation of how to interpret these ratios to aid in understanding the results. Overall, the study provides valuable insights into the relationship between VDT and surgical outcomes in early-stage NSCLC patients, but further research is needed to confirm these findings and address the limitations of the study.

The paper contains several grammar issues that need to be addressed to improve readability. 

Round 2

Reviewer 1 Report

Dear Authors,

Thank you for the opportunity to analyze your modified article. All your efforts are hihlighted by the precisions you have mentionned. it’s a better well written, easy reading and interesting article. Congratulations to authors for this work.

Please find my comments in the joined document. 
